# Clinical impact of intraoperative hyperlactatemia during craniotomy

Diana Romano [1]*, Stacie Deiner[1,2,3], Anjali Cherukuri[4☉], Bernard Boateng[5☉], Raj Shrivastava[2‡], J. Mocco[2‡], Constantinos Hadjipanayis[2‡], Raymund Yong[2‡], Christopher Kellner[2‡], Kurt Yaeger[2], Hung-Mo Lin[6], Jess Brallier[7]

1 Department of Anesthesiology, Perioperative and Pain Medicine, Icahn School of Medicine at Mount Sinai, New York, NY, United States of America, 2 Department of Neurosurgery, Icahn School of Medicine at Mount Sinai, New York, NY, United States of America, 3 Department of Geriatrics and Palliative Care, Icahn School of Medicine at Mount Sinai, New York, NY, United States of America, 4 University of Central Florida College of Medicine, Orlando, FL, United States of America, 5 Alabama College of Osteopathic Medicine, Dothan, AL, United States of America, 6 Department of Population Health Science and Policy, Icahn School of Medicine at Mount Sinai, New York, NY, United States of America, 7 Department of Anesthesiology and Critical Care Medicine, Memorial Sloan Kettering Cancer Center, New York, NY, United States of America

☉ These authors contributed equally to this work.
‡ These authors also contributed equally to this work.
* diana.romano@mountsinai.org

## Abstract

### Object

Patients often develop markedly elevated serum lactate levels during craniotomy although the reason for this is not entirely understood. Elevated lactate levels have been associated with poor outcomes in critically ill septic shock patients, as well as patients undergoing abdominal and cardiac surgeries. We investigated whether elevated lactate in craniotomy patients is associated with neurologic complications (new neurological deficits) as well as systemic complications.

### Methods

We performed a cohort study of elective craniotomy patients. Demographic and intraoperative data were collected, as well as three timed intraoperative arterial lactate values. Additional lactate, creatinine and troponin values were collected immediately postoperatively as well as 12 and 24 hours postoperatively. Assessment for neurologic deficit was performed at 6 hours and 2 weeks postoperatively. Hospital length-of-stay and 30-day mortality were collected.

### Results

Interim analysis of 81 patients showed that no patient had postoperative myocardial infarction, renal failure, or mortality within 30 days of surgery. There was no difference in the incidence of new neurologic deficit in patients with or without elevated lactate (10/26, 38.5% vs. 15/55 27.3%, p = 0.31). Median length of stay was significantly longer in patients with elevated lactate (6.5 vs. 3 days, p = 0.003). Study enrollment was terminated early due to futility (futility index 0.16).

**Data Availability Statement:** The data underlying the study has been deposited into Figshare and can be accessed there under the DOI 10.6084/m9. figshare.9955268.

**Funding:** The authors received no specific funding for this work.

**Competing interests:** The authors have declared that no competing interests exist.

## Conclusion

Elevated intraoperative serum lactate was not associated with new postoperative neurologic deficits, other end organ events, or 30 day mortality. Serum lactate was related to longer hospital stay.

## Introduction

In the United States, hundreds of thousands of craniotomies are performed every year.[1] Patients often develop markedly elevated intraoperative serum lactate levels, though the reason for this is not entirely understood.[2,3] Lactate, a substrate of carbohydrate metabolism, is formed from pyruvate during glycolysis. Its production reflects the magnitude of anaerobic metabolism that is related to cellular hypoxia[4,5] and it is encountered in a multitude of clinical presentations and disease states.[6] Elevated lactate levels have been closely related to poor outcomes in critically ill septic shock patients,[7–10] as well as patients undergoing abdominal [11] and cardiac surgeries.[12,13]

The etiology and clinical importance of elevated lactate during craniotomy can be difficult to determine. Patients undergoing craniotomy are at risk for developing systemic end-organ hypoperfusion secondary to fluid under resuscitation, hemorrhage, comorbid medical conditions (sepsis, congestive heart failure), or a combination of these factors. However, in some cases, elevated serum lactate could reflect localized changes in cerebral metabolism caused by perturbations such as brain retraction and tumor burden.[4,14,15] Deciding whether hyperlactatemia is due primarily to systemic versus local factors is essential because the management strategies differ. More important, however, is determining whether hyperlactatemia is associated with clinically significant patient outcomes in this patient population. This would help determine whether rising lactate values on intraoperative blood gas analysis are worth attempting to treat.

To assess the clinical significance of elevated serum lactate in the craniotomy population, our group previously performed a retrospective study comparing outcomes between those with and without elevated intraoperative serum lactate.[3] Elevated intraoperative serum lactate was associated with an increased risk of developing a new postoperative neurological deficit and longer length of hospital stay, but not an increased risk of renal failure, myocardial infarction, or mortality. In order to validate the findings from our previous report, we performed a prospective study. We hypothesized that in the craniotomy population, serum lactate is a marker of early regional hypoperfusion confined to the brain and that elevated intraoperative levels are associated with the development of new neurological deficits and longer hospital length of stay, but not associated with end-organ events such as myocardial infarction, renal failure or mortality. If elevated intraoperative serum lactate is associated with new neurologic deficit then rising serum lactate should be actionable, or at least serve as a warning sign for impending poor neurologic outcome.

## Materials and methods

Following approval by the Institutional Review Board of Mount Sinai Hospital (IRB-17-01480), informed consent was obtained from eligible patients between May 2017 and July 2018 at The Mount Sinai Hospital, New York, NY. Craniotomy cases were identified from the electronic operating room schedule prior to the day of surgery. We reviewed the patient's

electronic medical record (Epic Systems, Verona, WI) for inclusion and exclusion criteria, and interviewed the patient for confirmation. Inclusion criteria were: adult, English-speaking, and scheduled to undergo elective craniotomy for a brain mass or lesion involving the cranial nerves, neuroendocrine system, neurovascular system or dural-meningeal system. Exclusion criteria included cerebrospinal fluid shunt procedures, deep brain stimulation, reoperations during the same hospital admission, and cognitive impairment precluding consent. Patients with comorbid conditions that might cause increased lactate levels including pre-existing severe liver or kidney disease and sepsis were also excluded.

Preoperative data collected included date of service, age, gender, race, ethnicity, height, weight, BMI, ASA Physical Status Classification (as assigned by the anesthesiologist performing the case), preoperative diagnosis, anatomical location of the lesion, medical comorbidities (cardiovascular, respiratory, and other), baseline creatinine and presence or absence of preoperative neurologic deficit.

Anesthesiologists were given minimal instructions other than to avoid the use of Lactated Ringer's for intravenous fluids as this could hypothetically confound lactate measurements. As per our institutional standard of care for craniotomy, anesthesiologists placed arterial lines in all patients prior to procedure start. Arterial blood gas samples including lactate levels were obtained by the anesthesiologists from the arterial line at three different points during the procedure: 1) baseline prior to craniotomy, 2) after craniotomy during surgical manipulation of the brain, and 3) during surgical closure. All arterial blood gas values were measured on GEM-STAT 3000 Premier (Instrumentation Laboratories, Bedford, MA). Patients were required to have these three arterial blood gas values at minimum; however, anesthesiologists were free to draw additional arterial blood gas values at their discretion. Elevated lactate was defined as any value greater than or equal to 2 mmol/L based on the upper limit of normal for our institution's laboratory and on review of cardiac and general surgery literature. Other intraoperative data collected included procedure, positioning, total intravenous fluids, blood products, estimated blood loss, urine output, mannitol administration (or not), total surgical time, and whether or not the patient was extubated at the end of the procedure.

The primary outcome of new neurological deficit was assessed at two time points: 1) by neurosurgical house staff approximately 6 hours postoperatively in accordance with their usual postsurgical neurologic assessment timeline and 2) at the patient's follow up visit with their neurosurgeon approximately 2 weeks postoperatively. A complete neurological examination was performed by neurosurgeons to isolate postoperative neurologic changes. This included the absence or presence of disorientation, dysarthria, aphasia, motor weakness, hemiplegia, dysmetria, pronator drift, sensory deficits, anisocoria, abnormal eye movements, facial droop, tongue deviation, hearing deficit, visual acuity or field deficit, facial sensory deficit/alteration, or events such as seizures or autonomic dysfunction.

Secondary outcomes were assessed through collection of immediate postoperative lactate, immediate postoperative troponin, immediate postoperative creatinine, 12–24 hour postoperative lactate, 12–24 hour postoperative creatinine, hospital length of stay, in-hospital mortality, and 30 day postsurgical mortality by chart review. All data were entered into a Research Electronic Data Capture tool (REDCap; Vanderbilt University, Nashville, TN) for data analysis.

## Sample size calculation

In our previously published retrospective study, new neurological deficit rates were 7.3% and 14.7% for patients with normal and elevated lactate respectively. The prevalence of elevated lactate was 48%. We estimated that a two group continuity corrected chi-squared test with a 0.05 two-sided significance level would have 80% power to detect a difference in neurologic

deficit between patients with and without elevated lactate with sample sizes of 323 and 294 respectively yielding a total sample size of 617. The planned study period was two years and interim analysis was performed one year into the study. The rates of new neurological deficit were 30% (15/50) and 37.5% (9/24), respectively, for patients with normal and elevated lactate. The difference in new neurological deficit between patients with normal and elevated lactate was unlikely to reach statistical significance (futility index 0.16) within our planned study period. The decision was then made to terminate patient enrollment (conditional power calculation based on PASS v 12.0.2, NCSS, LLC).

## Statistical analysis

Descriptive statistics are presented as mean (SD), median [quartile 1 to quartile 3], or N (%), as appropriate. Comparisons of outcomes (neurological deficit, in-hospital or 30-day mortality) by serum lactate condition (elevated or normal) were performed using chi-squared, Fisher-exact, 2-sample t, and Wilcoxon rank-sum tests where appropriate. Logistic regression modeling was used with backward selection (stay criterion of 0.1) to identify predictors of new postoperative neurologic deficit. Cox modeling was used to model time to hospital discharge with backward selection (stay criterion 0.1). Initially, we adjusted both models for gender, surgical time, ASA classification, BMI, age, presence of pre-operative neurologic deficit, intraoperative lactate $\geq 2$, and preoperative diagnosis (infratentorial or supratentorial lesion). The only outcome that was investigated in the interim analysis was new neurologic deficit. We did not further adjust for its p-value in the final analysis, as it does not change the conclusion of the finding. Analyses were completed using SAS software 9.4 (SAS Institute).

## Results

A total of 85 patients were enrolled over the study period (May 2017-July 2018). Three patients were excluded due to missing intraoperative lactate values, and one patient was excluded due to surgery cancellation prior to induction of anesthesia (Fig 1).

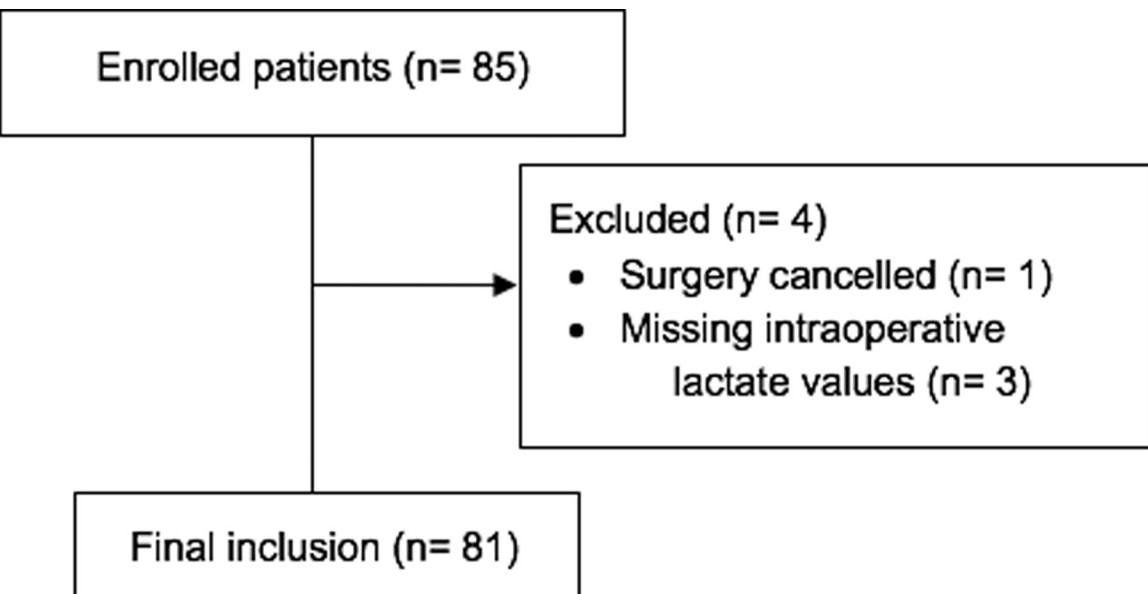

**Fig 1. Flow diagram.** Flow diagram of enrolled, excluded and included patients.

Table 1 shows the baseline characteristics of the patient cohort, both with and without elevated lactate. There was no significant difference between patients with and without elevated intraoperative lactate with regard to age, gender, race, ethnicity, ASA physical status classification, BMI, location of procedure (supratentorial versus infratentorial), comorbidities, or the presence of preoperative neurologic deficit. Patients with elevated lactate received more IV fluids intraoperatively and had greater estimated blood loss, without significant differences in blood products, urine output, total surgical time, mannitol administration, or extubation status.

Table 2 compares postoperative outcomes in patients with and without elevated intraoperative lactate. No patient had evidence of postoperative myocardial infarction or renal failure and there were no mortalities within 30 days of surgery. There was no statistically significant difference in the incidence of new neurologic deficits in patients with or without elevated lactate (10/26, 38.5% vs. 15/55 27.3%, p = 0.31). Patients with a new neurologic deficit had a median of the average intraoperative lactate of 1.27 [IQR 1.0–1.9], compared to patients without a new neurologic deficit 1.08 [IQR 0.85–1.70] p = 0.39. Median length of stay was significantly longer in patients with elevated intraoperative lactate versus without elevated lactate (6.5 vs. 3 days, p = 0.003).

Next, possible predictors of new neurologic deficit including age, gender, BMI, ASA classification, length of surgery, estimated blood loss, preoperative diagnosis (location of lesion), and presence or absence of preoperative neurologic deficit were examined using logistic regression analysis (Table 3). Elevated intraoperative serum lactate was not found to be an independent predictor of postoperative neurologic deficit in this model (p = 0.23). The only variable found with a trend toward significance for new postoperative neurologic deficit was location of the lesion; the odds of new deficit in patients with infratentorial lesions was 2.5 times that of patients with supratentorial lesions (95% CI: 0.94–6.64; p = 0.067).

Lastly, length of stay was modeled using Cox proportional hazards modeling (Table 4). Variables included were the same as in Table 3. Predictors associated with hospital length of stay were BMI and intraoperative lactate ≥2. Patients with higher BMI were more likely to be discharged sooner (relative discharge probability 1.08; CI 1.05–1.13; p <0.001). Patients with elevated intraoperative lactate were less likely to be discharged (relative discharge probability 0.59; 95% CI 0.37–0.97; p = 0.036).

## Discussion

We found that patients with and without elevated lactate had no difference in the incidence of new neurologic deficit 2 weeks postoperatively. Differences between the groups included longer length of stay in the elevated lactate group, with elevated lactate as a predictor of lengthened hospital stay.

Longer hospital stay in those with elevated intraoperative lactates is a finding consistent with our retrospective study.[3] While the two groups were similar in medical comorbidities, it is possible that elevated lactate may be a sign of systemic hypoperfusion in cases that involved fluid shifts and/or changes in perfusion. Furthermore, logistic regression analysis revealed elevated lactate to predict longer length of stay. Reasons for this are not readily apparent, however there may be postoperative variables other than a neurologic deficit that could possibly explain this, such as infection, DVT, subclinical cerebral ischemia that would manifest as diffusion restriction on post-operative MRI, or other reasons that could be measured in future studies to validate or better explain this association.

Our results differ from our retrospective study with regard to the finding of new neurologic deficit.[3] It is possible that prospective screening for new neurologic deficits using certain

**Table 1. Characteristics of patient cohort, grouped by level of maximum intraoperative serum lactate measurement.**

| Variable | | Overall | Max lactate < 2 (N = 55) | Max lactate ≥ 2 (N = 26) | P |
|---|---|---|---|---|---|
| Age | | 50.6 (14.0) | 51.6 (13.6) | 48.5 (14.9) | 0.36 |
| Gender | | | | | 0.28 |
| | Male | 46 (56.8) | 29 (52.7) | 17 (65.4) | |
| | Female | 35 (43.2) | 26 (47.3) | 9 (34.6) | |
| Race | | | | | 0.41 |
| | American Indian/Alaska Native | 0 | 0 | 0 | |
| | Asian | 8 (9.9) | 7 (12.7) | 1 (3.9) | |
| | Native Hawaiian/Pacific Islander | 0 | 0 | 0 | |
| | Black or African American | 14 (17.3) | 11 (20) | 3 (11.5) | |
| | White | 55 (67.9) | 34 (61.8) | 21 (80.8) | |
| | Unknown | 4 (4.9) | 3 (5.5) | 1 (3.8) | |
| Ethnicity | | | | | 0.67 |
| | Hispanic or Latino | 13 (16) | 8 (14.6) | 5 (19.2) | |
| | Not Hispanic or Latino | 66 (81.5) | 46 (83.6) | 20 (76.9) | |
| | Unknown | 2 (2.5) | 1 (1.8) | 1 (3.8) | |
| ASA PS | | | | | 0.61 |
| | 1–2 | 28 (34.6) | 21 (38.2) | 7 (26.9) | |
| | 3 | 49 (60.5) | 31 (56.4) | 18 (69.2) | |
| | 4 | 4 (4.9) | 3 (5.4) | 1 (3.8) | |
| BMI | | 27.3 (6.2) | 27.4 (6.6) | 27.1 (5.3) | 0.82 |
| Location of lesion | | | | | 0.64 |
| | Supratentorial | 50 (61.7) | 33 (60) | 17 (65.4) | |
| | Infratentorial | 31 (38.3) | 22 (40) | 9 (34.6) | |
| Cardiovascular disease | | | | | |
| | Hypertension | 32 (39.5) | 22 (40) | 10 (38.5) | 0.90 |
| | CAD | 3 (3.7) | 3 (5.4) | 0 | 0.55 |
| | CHF | 0 | 0 | 0 | |
| | Other | 53 (65.4) | 37 (67.3) | 16 (61.5) | 0.61 |
| Respiratory disease | | | | | |
| | Asthma | 8 (9.9) | 6 (10.9) | 2 (7.7) | 1.00 |
| | COPD | 4 (4.9) | 2 (3.6) | 2 (7.7) | 0.59 |
| | Sleep apnea | 6 (7.4) | 4 (7.3) | 2 (7.7) | 1.00 |
| | Other | 67 (82.7) | 46 (83.6) | 21 (80.8) | 0.76 |
| Other PMH: | | | | | |
| | Diabetes | 11 (13.6) | 10 (18.2) | 1 (3.8) | 0.10 |
| | Renal | 1 (1.2) | 0 | 1 (3.8) | 0.32 |
| | Hepatic | 4 (4.9) | 2 (3.6) | 2 (7.7) | 0.59 |
| | Other | 72 (88.9) | 48 (87.3) | 24 (92.3) | 0.71 |
| Pre-op neuro deficit present? | | | | | 0.10 |
| | Yes | 59 (72.8) | 37 (67.3) | 22 (84.6) | |
| | No | 22 (27.2) | 18 (32.7) | 4 (15.4) | |
| Total IV fluids | | 2300 (1700–3200) | 2000 (1500–2500) | 3125 (2000–4000) | **0.006** |
| Total blood products | | 0 (0–0) | 0 (0–0) | 0 (0–0) | 0.14 |
| Total EBL | | 150 (100–250) | 150 (100–200) | 200 (150–400) | **0.012** |
| Total urine output | | 1775 (1200–2300) | 1700 (1100–2200) | 2050 (1350–2900) | 0.07 |
| Mannitol given (y/n) | | 75 (92.6) | 51 (92.7) | 24 (92.3) | 1.00 |
| Total surgical time (min) | | 223 (167–306) | 217 (162–282) | 258 (172–417) | 0.17 |

*(Continued)*

**Table 1.** (Continued)

| Variable | Overall | Max lactate < 2 (N = 55) | Max lactate ≥ 2 (N = 26) | P |
|---|---|---|---|---|
| Patient extubated end of case | | | | 0.20 |
| Yes | 74 (91.4) | 52 (94.6) | 22 (84.6) | |
| No | 7 (8.6) | 3 (5.4) | 4 (15.4) | |

Note: Continuous variables are given mean (SD) with t test or median (Q1-Q3) with Wilcoxon rank test. Categorical variables are given N (col%) with chi-square test.

specific criteria yielded different results than retrospective chart review. Assessment of neurologic status in the current study is more comprehensive than in the retrospective study[3], with assessments being performed at standardized time intervals and using certain common deficits as delineated to us by our neurosurgical colleagues (KY). Next, our current study recruited outpatient elective craniotomy patients whereas the previous study sample was found by CPT code for craniotomy and may have included a wider variety of patients, such as inpatients, emergency surgeries, patients with altered mental status, and moribund patients that would have been excluded from the current study. Furthermore, sensitivity analysis in the retrospective study comparing patients in the study cohort to those excluded due to no lactate levels drawn revealed those patients without lactates drawn to be significantly healthier. In our current study cohort, all patients, by design, had intraoperative lactates drawn. As these were elective outpatient craniotomy cases, it is likely that the anesthesiologists may not have otherwise drawn ABGs on these patients intraoperatively, and thus a large number of this type of patient would have been excluded from the previous study. This supports that the study populations were likely somewhat different between the two studies and may have led to the difference in the finding of new neurologic deficit.

One reason that we may not have found a difference between groups with regard to new neurologic deficit is that the contribution of surgical manipulation to overall systemic lactate levels may not be measurable. Other studies have used microdialysis catheters to study cerebral

**Table 2. Postoperative outcomes with and without elevated intraoperative serum lactate.**

| Outcome | N* | Overall | Max lactate < 2 (N = 55) | Max lactate ≥ 2 (N = 26) | P |
|---|---|---|---|---|---|
| Postoperative troponin > 0.05 ng/mL | 75 | 0 | 0 | 0 | |
| Postoperative creatinine > 1.10 mg/dL | | | | | |
| Immediate post op | 81 | 3 (3.7) | 2 (3.6) | 1 (3.8) | 1.00 |
| 12–24 hours post op | 77 | 3 (3.9) | 2 (3.9) | 1 (3.8) | 1.00 |
| Postoperative lactate ≥2 | | | | | |
| Immediate postoperative | 78 | 45 (57.7) | 22 (42.3) | 23 (88.5) | **<0.001** |
| 12–24 hours postoperative | 68 | 38 (55.9) | 21 (45.7) | 17 (77.3) | **0.01** |
| Immediate postoperative neuro deficit | 81 | | | | 0.55 |
| Yes | | 35 (43.2) | 25 (45.5) | 10 (38.5) | |
| No | | 46 (56.8) | 30 (54.5) | 16 (61.5) | |
| Hospital length of stay | 81 | 4 (3–7) | 3 (3–6) | 6.5 (3–7) | **0.003** |
| New neurologic deficit at 2 week follow up | 81 | | | | **0.309** |
| Yes | | 25 (30.9) | 15 (27.3) | 10 (38.5) | |
| No | | 56 (69.1) | 40 (72.7) | 16 (61.5) | |
| 30 day mortality | 81 | 0 | 0 | 0 | |

Note: Continuous variables are given mean (SD) with t test or median (Q1-Q3) with Wilcoxon rank test. Categorical variables are given N(col%) with chi-square test.

* = number of patients with available data.

**Table 3. Logistic model of factors associated with new neurological deficit.**

| Factor | Odds Ratio | 95% Confidence Interval | *P* |
|---|---|---|---|
| Age (y) | | | 0.51 |
| Female | | | 0.20 |
| BMI (kg/m$^2$) | | | 0.97 |
| ASA Classification | | | 0.90 |
| Preoperative neurologic deficit present | | | 0.34 |
| Surgical time | | | 0.99 |
| Infratentorial craniotomy | 2.495 | 0.937–6.664 | 0.07 |
| EBL | | | 0.54 |
| Lactate $\geq$ 2 mmol/L | | | 0.23 |

metabolites in patients undergoing meningioma resections and found that certain metabolites, namely lactate, pyruvate, and glucose, can change with the use of temporary arterial occlusion for proximal control during intracranial aneurysm surgery. One study found that a higher pyruvate: lactate ratio postoperatively was associated with unfavorable outcomes in patients.[16] Whether or not the lactate found in cerebral microdialysis catheters can be extrapolated to serum lactate remains to be seen. On the other hand, transcranial magnetic stimulation of the brain has been shown to increase serum lactate lending support to the notion that brain lactate production may indeed be linked to that of the serum. [17] Along similar lines, studies have suggested cerebral hyperlactatemia as a potential biomarker for malignancy of brain tumors.[14,15]

Studies have pointed to the questionable clinical importance of elevated serum lactate in neurosurgical patients. One such study examined postoperative serum lactate levels in a neurosurgical ICU including brain tumor, spinal surgery and non-neurosurgical patients and found higher serum lactate levels in brain tumor and spinal surgery patients compared to non-neurosurgical patients. The primary outcome was mortality and the lactates were obtained postoperatively rather than intraoperatively. They found that elevated serum lactate was not associated with mortality.[2] Other studies have shown elevated serum lactate levels both preoperatively and intraoperatively not to be associated with mortality in patients undergoing craniotomies or craniectomies.[18,19] Elevated serum lactate levels have also been attributed to administration of furosemide with mannitol which is common in neurosurgery.[20] Additionally, in both our current and previous study, no patient had evidence of postoperative end-organ dysfunction such as MI, renal failure or death within 30 days of surgery. In the current study, given the small number of patients enrolled and that catastrophic outcomes such as mortality and end organ dysfunction were relatively rare, this particular result is expected.

**Table 4. Survival model of hospital length of stay (days).**

| Factor | Hazard Ratio | 95% Confidence Interval | *P* |
|---|---|---|---|
| Age (y) | | | 0.90 |
| Female | | | 0.53 |
| BMI (kg/m$^2$) | 1.084 | 1.045–1.125 | **<0.001** |
| ASA Classification | | | 0.29 |
| Preoperative neurologic deficit present | | | 0.90 |
| Surgical time | | | 1.16 |
| Infratentorial craniotomy | | | 0.99 |
| EBL | | | 0.44 |
| Lactate $\geq$ 2 mmol/L | 0.594 | 0.365–0.966 | **0.04** |

Interestingly, in contrast to the association of lactate with unfavorable outcome, there is a growing body of literature regarding lactate as protective or beneficial in brain injury. Studies have postulated that during brain stress the body may release stores of lactate to feed into the TCA cycle of neurons (the astrocyte-neuron lactate shuttle hypothesis).[21] Brain uptake of systemic lactate "may reflect an adaptive response to the increased energy demands and change in metabolic priorities of the injured brain."[21] Endogenous lactate use by the injured brain implies that administering lactate to traumatic brain injury patients could be advantageous, and studies of rats given intravenous lactate after fluid percussion injury to the brain actually performed better than those given saline.[22,23] This brings up a provocative question–Is the lactate routinely measured intraoperatively during brain surgery coming from the brain due to surgical manipulation, or is it instead being mobilized from the body to serve the brain for assistance with the injury/stress of surgery?

In our study we observed a trend that patients with infratentorial lesions were 2.5 times more likely to have a new postoperative neurologic deficit than patients with supratentorial lesions (p = 0.067). One other study found that due to the anatomical vicinity to delicate brain structures and the relatively narrower surgical corridors employed during posterior fossa surgery, arteriovenous malformations in the infratentorial region are more hazardous and prone to rupture leading to worse neurological outcome compared to those that were supratentorial. [24] We did not separate our findings by type of surgery; however, it is clear that further research is needed to elucidate whether location of the lesion is associated with new neurologic deficit.

In this cohort, patients with higher BMI were more likely to be discharged sooner (relative discharge probability 1.08). Overall our patient population had a median BMI of 27.3 with a standard deviation of 6.2. One possible explanation for the findings regarding BMI is that the lower BMI patients were either frail or cachectic (possibly due to malignancy) and might have had less physiologic reserve and required greater recuperation time post surgically whereas a normal or slightly elevated BMI may have been protective, giving the patient higher likelihood of early hospital discharge. The "obesity paradox" suggests that, counterintuitively, excess body weight may be protective and associated with improved outcomes in certain groups. Studies have confirmed the presence of this paradox in patients with neurologic injury. Specifically, studies of patients with ischemic and hemorrhagic stroke showed overweight or mildly obese status conferred greater survival.[25,26] While these studies focused on stroke patients rather than post-surgical patients, at least one study showed shorter hospital length of stay in patients with higher BMI.[27]

Lastly, similar to our previous study, patients with elevated serum lactate received more intravenous fluids intraoperatively than their normal lactate counterparts (3125mL vs 2000mL; p = 0.006). This suggests that anesthesiologists tended to respond to or attempt to "treat" rising lactate by administering additional fluids. It is possible that the elevated lactate may have been due to under-resuscitation or cerebral hypoperfusion, the former but not the latter being corrected by additional fluid administration. This being said, neither systemic nor local hypoperfusion intraoperatively was associated with new postoperative neurologic deficits. Thus, one may speculate that it was likely temporary and/or reversible, and that the elevated lactate was not an important marker of impending poor neurologic outcome.

The major limitation of this study is that the study was terminated early due to high futility index in interim analysis. Although a small increase in the rate of new neurologic deficit was seen in patients with elevated lactate, this study did not demonstrate a clinically significant difference between the two groups. Future prospective studies with larger sample size could eliminate the possibility of a beta error. Similar to our retrospective study[3], we used systemic markers such as serum lactate without correlation to direct measures of cerebral perfusion

such as use of jugular bulb sampling. Similarly, data such as cerebral blood flow or NIRS measurements were not available.

## Conclusions

In conclusion, in this prospective study of patients undergoing craniotomy, we sought to determine whether elevated lactate in craniotomy patients was associated with neurologic as well as systemic complications. On interim analysis, elevated intraoperative lactate level was not independently associated with new postoperative neurologic deficits, renal failure, myocardial infarction, or 30 day mortality. Patients with elevated lactate had significantly longer hospital admissions, with the potential for increased hospital costs. The difference in incidence of new neurologic deficit between patients with and without elevated intraoperative serum lactate was small, with a high futility index thus enrollment was terminated. While this study includes a relatively small number of patients, the results indicate that serum lactate is unlikely to be a marker of poor neurologic outcome and isolated hyperlactatemia (without signs of systemic compromise such as bleeding, etc.) might not be a cause for alarm when encountered intraoperatively in craniotomy patients. Future, larger-scale prospective studies can focus on different markers such as cerebral lactate or other measures of cerebral perfusion and assess whether these are of more value to intraoperative anesthetic and surgical decision making.

## Author Contributions

**Conceptualization:** Diana Romano, Stacie Deiner, Hung-Mo Lin, Jess Brallier.

**Data curation:** Diana Romano, Anjali Cherukuri, Bernard Boateng.

**Formal analysis:** Diana Romano, Hung-Mo Lin.

**Investigation:** Diana Romano, Raj Shrivastava, J. Mocco, Constantinos Hadjipanayis, Raymund Yong, Christopher Kellner, Kurt Yaeger.

**Methodology:** Diana Romano, Stacie Deiner, Kurt Yaeger, Jess Brallier.

**Project administration:** Diana Romano, Jess Brallier.

**Resources:** Diana Romano, Raj Shrivastava, J. Mocco, Constantinos Hadjipanayis, Raymund Yong, Christopher Kellner, Kurt Yaeger.

**Software:** Diana Romano.

**Supervision:** Stacie Deiner, Jess Brallier.

**Validation:** Hung-Mo Lin.

**Writing – original draft:** Diana Romano.

**Writing – review & editing:** Diana Romano, Stacie Deiner, Raj Shrivastava, J. Mocco, Constantinos Hadjipanayis, Raymund Yong, Christopher Kellner, Kurt Yaeger, Hung-Mo Lin, Jess Brallier.

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
