## [Decision Letter · Decision Letter 0]

4 Oct 2019

Clinical impact of intraoperative hyperlactatemia during craniotomy

PONE-D-19-23758

Dear Dr. Romano ,

We are pleased to inform you that your manuscript has been judged scientifically suitable for publication and will be formally accepted for publication once it complies with all outstanding technical requirements.

With kind regards,

Ehab Farag, MD FRCA FASA

Academic Editor

PLOS ONE

**Journal Requirements**

**Comments to the Author**

1. Is the manuscript technically sound, and do the data support the conclusions?

Reviewer #1: Yes

Reviewer #2: Yes

2. Has the statistical analysis been performed appropriately and rigorously? 

Reviewer #1: Yes

Reviewer #2: Yes

3. Have the authors made all data underlying the findings in their manuscript fully available?

Reviewer #1: Yes

Reviewer #2: Yes

4. Is the manuscript presented in an intelligible fashion and written in standard English?

Reviewer #1: Yes

Reviewer #2: Yes

5. Review Comments to the Author

Reviewer #1: Romano et al present a prospective cohort analysis of the role serum lactate in craniotomy patients. This is a well written well designed study that is a continuation of their previous retrospective analysis. Listed below are some constructive suggestions.

1. Would like to see a few paragraphs describing how the prospective analysis design compares to the retrospective analysis. In the prospective design there appears to be a defined regimen in terms of frequency of ABG, tropinin and creatinine draws, neuro checks, and details of the neuro exam. But in a retrospective review it is common to find out certain regimens may not have been instituted. For instance, it appears that in the retrospective patients there may have been a lack of consistency in the frequency of ABG draws. I would like to know what were the limitations of the retrospective design and how that could have led to different findings between the retrospective and prospective studies.

2. Would like an explanation on why base deficit or base excess results were not presented. Would like a commentary on whether or not base deficit measurement would represent a good correlate or surrogate marker in relation to lactate for brain perfusion and end organ tissue perfusion. Also further commentary on administering albumin vs lactated ringers vs normal saline on their effects on lactate, base deficit and tissue oxygenation.

3. The authors also commented on the role of microdialysis in meningioma patients. Will there be future research plans to compare serum lactate in relation to cerebral microdialysis glucose and lactate/pyruvate ratios (Carteron et al, Front Neurol. 2017; 8: 601). Also would like to have commentary on how serum or cerebral lactate could differ in the setting of craniotomy for tumor excision vs craniotomy/embolization of neurovascular lesions (aneurysms, avm)

Reviewer #2: Rosenberger Review: Manuscript PONE-D-19-23758

1. What are the main claims of the paper and how significant are they for the discipline?

→ Main purpose of the paper was to identify the significance of hyperlactatemia during craniotomy in regards to postoperative morbidity and mortality in a prospective randomized trial and also its critical differential diagnosis for the clinican

2. Are the claims properly placed in the context of the previous literature? Have the authors treated the literature fairly?

→ Literature was treated fairly. The authors published a retrospective study showing clinical significance of intraoperative hyperlactatemia for postoperative outcome and therefore decided to follow up with a prospective trial

3. Do the data and analyses fully support the claims? If not, what other evidence is required?

→ Data analyses revealed no significant impact on patient outcome after study was terminated early for high futility index in interim analysis.

4. PLOS ONE encourages authors to publish detailed protocols and algorithms as supporting information online. Do any particular methods used in the manuscript warrant such treatment? If a protocol is already provided, for example for a randomized controlled trial, are there any important deviations from it? If so, have the authors explained adequately why the deviations occurred?

→ Authors did follow a detailed protocol, all methods were addressed, no deviations apparent for reviewer. The study was terminated early due to high futility index in interim analysis.

5. If the paper is considered unsuitable for publication in its present form, does the study itself show sufficient potential that the authors should be encouraged to resubmit a revised version?

→ I recommend considering a publication in the current format based on the fact that the authors published a retrospective study and followed scientific principles to validate their previously published retrospective results with a prospective trial. The discussion includes reasons why retrospective results differ from the prospective study results and this message is important to be shared. It allows the reader to critically review results published based on retrospective data analysis versus prospective analysis - and how it can change clinical practice management if data is based on retrospective data only. The discussion also addresses the difference of serum lactate as a biomarker in neurosurgical patients versus the understanding and differential diagnosis of elevated serum lactate in the general critical care patient population. This is of value for critical care and non-critical care specialists.

6. Are original data deposited in appropriate repositories and accession/version numbers provided for genes, proteins, mutants, diseases, etc.?

→ Yes

7. Are details of the methodology sufficient to allow the experiments to be reproduced?

→ Yes

8. Is the manuscript well organized and written clearly enough to be accessible to non-specialists?

→ Yes. I do compliment the research team for their candid data review. Thorough analysis of their own data from a retrospective study going forward to prospectively validate the results and critically assess further hypothesis why the current hypothesis of hyperlactatemia in craniotomy did not impact neurologic outcome is in clear and concise form, and accessible to non-specialists.

6. PLOS authors have the option to publish the peer review history of their article (what does this mean?). If published, this will include your full peer review and any attached files.

Reviewer #1: No

Reviewer #2: Yes: Dorothea S Rosenberger MD PhD

---

## [Editor Report · Acceptance letter]

11 Oct 2019

PONE-D-19-23758 

Clinical impact of intraoperative hyperlactatemia during craniotomy 

Dear Dr. Romano:

I am pleased to inform you that your manuscript has been deemed suitable for publication in PLOS ONE. Congratulations! Your manuscript is now with our production department. 

With kind regards,

on behalf of

Dr. Ehab Farag 

Academic Editor

PLOS ONE